# Application of 5-Methylcytosine DNA Glycosylase to the Quantitative Analysis of DNA Methylation

**DOI:** 10.3390/ijms22031072

**Published:** 2021-01-22

**Authors:** Woo Lee Choi, Young Geun Mok, Jin Hoe Huh

**Affiliations:** 1Department of Agriculture, Forestry and Bioresources, Seoul National University, Seoul 08826, Korea; woori8906@snu.ac.kr; 2Interdisciplinary Program in Agricultural Genomics, Seoul National University, Seoul 08826, Korea; ygmok@IBS.re.kr; 3Research Institute of Agriculture and Life Sciences, Seoul National University, Seoul 08826, Korea; 4Plant Genomics and Breeding Institute, Seoul National University, Seoul 08826, Korea

**Keywords:** DNA methylation, DEMETER, DNA demethylase, epiallele, epigenetic profiling

## Abstract

In higher eukaryotes DNA methylation is a prominent epigenetic mark important for chromatin structure and gene expression. Thus, profiling DNA methylation is important for predicting gene expressions associated with specific traits or diseases. DNA methylation is achieved by DNA methyltransferases and can be actively removed by specific enzymes in a replication-independent manner. DEMETER (DME) is a bifunctional 5-methylcytosine (5mC) DNA glycosylase responsible for active DNA demethylation that excises 5mC from DNA and cleaves a sugar-phosphate bond generating a single strand break (SSB). In this study, DME was used to analyze DNA methylation levels at specific epialleles accompanied with gain or loss of DNA methylation. DME treatment on genomic DNA generates SSBs in a nonsequence-specific fashion proportional to 5mC density, and thus DNA methylation levels can be easily measured when combined with the quantitative PCR (qPCR) method. The DME-qPCR analysis was applied to measure DNA methylation levels at the *FWA* gene in late-flowering *Arabidopsis* mutants and the *CNR* gene during fruit ripening in tomato. Differentially methylated epialleles were successfully distinguished corresponding to their expression levels and phenotypes. DME-qPCR is proven a simple yet effective method for quantitative DNA methylation analysis, providing advantages over current techniques based on methylation-sensitive restriction digestion.

## 1. Introduction

DNA methylation is a prominent epigenetic modification in higher eukaryotes crucial for many biological processes such as differentiation, gene imprinting, and X chromosome inactivation [1,2,3]. DNA methylation is generally referred to as the addition of a methyl group to cytosine producing 5-methylcytosine (5mC) in DNA. DNA methylation is established and maintained by de novo DNA methyltransferase and maintenance DNA methyltransferase, respectively, and can also be removed by a passive or an active mechanism in response to developmental cues. Passive DNA demethylation is replication-dependent, whereas active DNA demethylation is replication-independent and requires enzyme activity [2,3]. DNA methylation changes are often associated with alteration of genes expression causing various diseases and, notably cancer in humans. Therefore, changes in DNA methylation patterns of specific gene can serve as biomarkers in cancer diagnosis. For example, hypermethylation of *glutathione S-transferase P* (*GSTP1*) is a hallmark of prostate cancer in humans [4,5]. In plants, alteration of DNA methylation may contribute to changes in gene expression and phenotypic variations that can be sometimes transmitted to next generations. The inherited alleles that maintain differences in DNA methylation and expression patterns are referred to as epialleles [6], and several epialleles are reported to be associated with important traits in plants such as flower architecture, flowering time, fruit ripening, sex determination, starch metabolism, vitamin E accumulation, and oil productivity [7,8,9,10,11,12].

In DNA methylation investigation, DNA samples are often subjected to chemical or enzyme treatment prior to analysis depending on the analytical platform. Methylation-dependent pretreatments on genomic DNA were required to distinguish 5mC from unmethylated cytosine. Three main principles are utilized to distinguish methylated vs. unmethylated cytosines: (1) endonuclease digestion, (2) affinity enrichment, and (3) bisulfite conversion. First, endonuclease digestion relies on methylation-sensitive restriction endonucleases (MSREs) that are inhibited by 5mC in the recognition sequence, generating different cleavage patterns from the isoschizomers that recognize the same sequence. Second, affinity enrichment utilizes the methyl-binding protein such as MeCP2 and MBD2 [13,14] and immunoprecipitation with antibody against them. Alternatively, methylated genomic DNA can be immunoprecipitated with antibody specific for 5mC [15]. Third, sodium bisulfite treatment chemically converts only unmethylated cytosine residues, but not 5mC, to uracil, and therefore the methylation status can be distinguished at the base pair level [16].

Following the pretreatment, an array of techniques can be employed to delineate methylation profiles. Recent advances in high-throughput sequencing also allow genome-scale methylation profiling. In particular, whole-genome shotgun bisulphite sequencing (WGSBS) is achieved on an Illumina Genome Analyzer platform providing a single-base level high-resolution methylome profile in many eukaryote genomes, and thus regarded as a gold standard method at the moment [17]. In addition, BeadChip arrays which can detect a 5mC-specific “pseudo-SNP” through bisulfite conversion are popular as a low-cost alternative for genome-wide methylation profiling of a large number of samples, especially in humans [18].

Despite the benefits of bisulfite-based methods, enzyme-based approaches are still widely used for methylation analysis, because they provide a rapid, simple, cost-effective, and convenient platform that can be applied to a large sample size in many organisms [17,19]. MSREs such as *Hpa* II and *Sma* I are blocked by 5mC in their recognition sequences, whereas their isoschizomers *Msp* I and *Xma* I are not. Following digestion with MSREs, either gel electrophoresis combined with Southern hybridization or PCR amplification is widely used for DNA methylation analysis [20]. However, utilization of MSREs is highly restricted due mainly to specific recognition sequences that intrinsically limit the range of application. Besides MSREs, another enzyme-based technique employs a methylation-dependent homing endonuclease McrBC that recognizes and cleaves methylated DNA between two nonpalindromic G/A mC sites 40–3000 bp apart from each other [21]. The low sequence specificity of McrBC enables to reveal various DNA methylation patterns. Nevertheless, most enzyme-based methods are applied in practice to locus-specific analysis rather than genome-wide investigation.

Such enzyme-based techniques are substantially affected by the availability of recognition sequences, and thus DNA methylation analysis at any sequence context is not feasible. DEMETER (DME) is a bifunctional DNA glycosylase that directly recognizes and excises 5mC from DNA upon base excision via β- and δ-elimination in all sequence contexts; CG, CHG, and CHH (H = A, C or T) [22]. Unlike most DNA methylation-dependent restriction enzymes, DME acts in a sequence nonspecific manner [22]. Moreover, DME mechanistically permits a cleavage of only one strand of double-stranded DNA template at a symmetrically methylated site, generating a single strand break (SSB), rather than a double strand break (DSB) [22]. Therefore, DME-induced SSBs are proportional to the 5mC density within a methylated DNA fragment, and the quantification of 5mC-specific SSBs induced by DME treatment may overcome technical limitations of conventional enzyme-based methods, permitting the quantitative measurement of DNA methylation in all sequence contexts.

Here, we described a novel DNA methylation analysis technique to rapidly measure DNA methylation levels at specific loci. In the pretreatment step, DME excises 5mC and generates SSBs in proportion to the density DNA methylation, distinguishing methylated DNA from unmethylated DNA. The next step involves quantitative-PCR (qPCR) amplification of DME-treated template, whereby only under-methylated DNA fragments are more efficiently amplified. We demonstrated that the DME-qPCR method was sensitive and accurate enough to measure the ratio of methylated DNA vs. unmethylated DNA in heterogeneous samples. This technique was also successfully applied to the identification of hypomethylated *fwa* alleles in *Arabidopsis* according to the late flowering phenotype, and to monitoring changes in DNA methylation level at the *CNR* gene in tomato whose expression is affected by differentially methylated regions (DMRs) in the promoter during fruit ripening [8,9,23].

## 2. Results

### 2.1. DME Induces Single-Strand DNA Breaks Proportional to DNA Methylation Density

Most MSRE-dependent DNA methylation assays are rapid and cost-effective but have intrinsic limitations due largely to a narrow sequence specificity. First, we tested whether DME can distinguish unmethylated vs. methylated DNA using artificially methylated DNA fragments. Note that DME catalyzes 5mC excision generating SSBs at symmetrically methylated CG and CHG sequences, while DSB formation is intrinsically inhibited [22]. Genomic DNA isolated from *Saccharomyces cerevisiae* whose genome is devoid of DNA methylation [24] was in vitro methylated with *Hpa* II methyltransferase that methylates internal cytosine residues at the recognition sequence 5′-CCGG-3′. The *Hpa* II-methylated DNA was treated with DME and subjected to PCR amplification. The genomic region on yeast chromosome 1 (128,103–126,904) was PCR-amplified using the primer pairs that amplify the fragments with a single and three *Hpa* II sites, respectively (Figure 1A). The region on chromosome 7 (12,940–13,242) without an *Hpa* II sequence was chosen as an unmethylated control. The PCR amplification for the region with three *Hpa* II sites (with primers F1a and R1) was less than with a single site (with primers F1b and R1) (Figure 1B). This indicates that PCR amplification of DME-treated fragments is negatively proportional to the number of 5mCs in the target region. Additionally, in order to confirm that DME induced SSBs, rather than DSBs, DME-treated genomic DNA was digested with BAL-31 endonuclease, which induces DSBs at sites of SSBs. BAL-31 endonuclease incises the strand opposite the nick forming a DSB. We found that BAL-31 digestion of DME-treated fragments generated DSBs at methylated sites regardless of the number of 5mC bases (Figure 1C). This suggests that DME-induced SSB formation serves as an indicator of DNA methylation density in the region of interest, which can be measured by PCR amplification.

### 2.2. DME-qPCR Allows a Quantitative DNA Methylation Analysis

In order to assess the efficacy of DME treatment measuring the level of DNA methylation, we examined relative amplifications of DME-treated templates with different 5mC densities. The pUC19 plasmid was in vitro methylated with M.*Sss* I and *Hpa* II methyltransferases for methylation at CG and CCGG sequences, producing a heavily methylated region (HMR) with 20 5mCG and a low methylated region (LMR) with three 5mCG sites, respectively (Figure 2A). DME-induced SSB formation was compared with restriction by McrBC endonuclease, which is a type IV homing endonuclease cleaving one or two sites between two (A/T) 5mC sequences 40–2000 bp apart from each other. Both DME and McrBC treatments caused less PCR amplification for M.*Sss* I-methylated HMR than for the unmethylated region (UMR) (Figure 2B). DME treatment also caused less PCR amplification at the LMR, whereas McrBC treatment allowed amplification of both unmethylated and methylated templates to similar levels due to a lack of recognition sequences (Figure 2B). In order to assess the efficiency of DME treatment for quantitative measurement of DNA methylation in given samples, both unmethylated and M.*Sss* I-methylated plasmids were mixed in different ratios and subjected to DME treatment followed by quantitative real-time PCR (DME-qPCR) amplification. As shown in Figure 2C, the level of DME-qPCR amplification for HMR was proportional to the relative abundance of unmethylated template in the reaction. This allowed to quantitatively measure the ratio of methylated templates of the same sequence according to the equation fitted to the observed data (Figure 2C). These results strongly suggest that DME-qPCR can be utilized to analyze the level of DNA methylation at specific regions without concerning the sequence specificity that is regarded as one of the limitations of currently available MSRE-based methods.

### 2.3. DME-qPCR Distinguishes DNA Methylation Levels at the FWA Gene in Wild Type and Late Flowering Mutants in Arabidopsis

Epimutants display heritable changes in gene expression and phenotype associated with gain or loss of DNA methylation. *Arabidopsis fwa* mutants display a late-flowering phenotype caused by the loss of DNA methylation at the SINE elements proximal to the 5′-regulatory region [25]. Local bisulfite sequencing showed hypermethylation in the promoter region, particularly at CG sequences, of the *FWA* gene in early-flowering wild type (Figure 3A) but hypomethylation in late flowering *fwa* mutants (Figure 3B). We performed DME-qPCR for the *FWA* alleles in early-flowering Col-0 and L*er* wild-type accessions and late-flowering *fwa* mutants in the L*er* background. DME-qPCR produced significantly more amplification products for the hypomethylated *fwa* allele relative to hypermethylated *FWA* alleles of the wild type (Figure 3C), indicating that flowering time was determined by DNA methylation rather than by accession-specific genotypes. DME-qPCR enabled to successfully distinguish differences in DNA methylation between the wild type and *fwa* epimutants associated with flowering phenotypes.

We performed a cross between Col-0 wild type and L*er fwa-1* mutant to investigate whether DME-qPCR was also able to identify the epialleles associated with corresponding phenotypes in the segregating population. The F_1_ plant flowered late due to the dominant nature of *fwa* epimutation, and F_2_ individuals segregated for flowering time (early or late) (Figure 3D). DME-qPCR analysis successfully identified the hypermethylated or hypomethylated status of the *FWA* alleles in accordance with the time of flowering (Figure 3D). For instance, all individuals showing very low DME-qPCR amplification were early flowering and homozygous for *FWA*, whereas ones showing high amplification levels were found to carry at least one *fwa* allele with a late flowering phenotype (Figure 3D). Moreover, it is noteworthy that DME-qPCR was sensitive enough to distinguish between homozygous and heterozygous *fwa* mutants, where homozygous *fwa* mutants always produced more amplification signals than *fwa* heterozygotes (Figure 3D).

### 2.4. DME-qPCR Detects Changes in DNA Methylation Levels at the CNR Gene during Fruit Ripening in Tomato

The *Colorless non-ripening* (*Cnr*) mutant in tomato (*Solanum lycopersicum*) is a naturally occurring epimutant with a severe delay in fruit ripening, which is caused by gain of DNA methylation and transcriptional silencing of the *CNR* gene [9]. Bisulfite sequencing revealed that the upstream region of *CNR* gene of normal ripening ‘Ailsa Craig’ cultivar was nearly devoid of DNA methylation at any sequence context (Figure 4A), whereas the same region of non-ripening *Cnr* mutant was heavily methylated at CG and CHH sequences (Figure 4B). DME-qPCR also produced a high level of amplification for ‘Ailsa Craig’ *CNR* gene compared to the mutant *Cnr*, indicating that the *Cnr* epiallele is hypermethylated (Figure 4C). It was reported that two differentially methylated regions (DMRs) located at 0.7 and 1.9 kb upstream of the translation start site of *CNR* underwent gradual loss of DNA methylation during fruit ripening allowing transcriptional activation [23]. We harvested tomato fruits at different developmental stages and examined expression levels of *CNR* in the fruit pericarp of ‘Ailsa Craig’. Expression of *CNR* gradually increased until 42 days post-anthesis (d.p.a.) and decreased at 52 d.p.a. (Figure 4D). However, expression levels remained relatively low during the entire course of fruit ripening in *Cnr* mutants (Figure 4D). This is consistent with previous study showing *Cnr* expression patterns in wild type and *Cnr* mutant during tomato fruit ripening [9]. Accordingly, bisulfite sequencing revealed that DNA methylation at the −0.7 kb DMR of *CNR* progressively decreased in ripening ‘Ailsa Craig’ fruits but remained at high level in *Cnr* mutant fruits (Figure 4E). DME-qPCR analysis also verified dynamic changes in DNA methylation level for wild-type *CNR* during ripening (Figure 4F). In contrast, DME-qPCR amplification was constantly low in *Cnr* mutant fruits, indicating that the DMR retained a high level of DNA methylation responsible for transcriptional silencing and non-ripening phenotype (Figure 4F). These data suggest that DME-qPCR is highly applicable to monitoring DNA methylation changes at the gene of interest during developmental processes.

## 3. Discussion

Despite the benefits of using high-throughput technologies, enzyme-based DNA methylation analysis is still widely used because it offers a rapid, simple, and cost-effective solution. Current enzyme-based techniques mostly rely on the unique properties of MSREs that recognize but do not digest methylated CpG sites within recognition sequences. Therefore, use of MSREs, along with methylation-insensitive isoschizomers, provides unique restriction patterns reflecting the presence or density of DNA methylation, which can be readily detected by PCR amplification, electrophoresis, or gel blot analysis. However, application of MSREs is restricted to genomic regions that harbor recognition sequences, and thus, sequence coverage or resolution is relatively low. In addition, most MSREs such as *Hha* I, *Hpa* II, and *Sma* I and corresponding isoschizomers generate DSBs as a hallmark to indicate the presence or absence of 5mC, and therefore, quantitative analysis relying on DSB formation is impractical for densely methylated regions with multiple recognition sites or ones without it at all.

DME was first identified in *Arabidopsis* as a bifunctional 5mC DNA glycosylase [22,27]. DME removes a 5mC base in a sequence nonspecific manner by cleaving an N-glycosylic bond between 5mC and a ribose sugar, and accompanied lyase activity generates an SSB via β- and δ-elimination processes [22,28]. Notably, DME only induces an SSB at symmetrically methylated sites preventing DSB formation [22]. Therefore, DME-induced SSBs are formed in proportion to the 5mC density at methylated fragments, and an intact single DNA strand that still persists can serve as a template for PCR amplification.

In this study, we propose DME-qPCR as an alternative method to quantify DNA methylation density while overcoming the limits of current MSRE-based DNA methylation analyses. The advantages of DME-qPCR over MSRE-dependent techniques may include sequence nonspecificity, better sensitivity that allows quantitative analysis, and a higher resolution it provides. Since DME is able to excise 5mC from both symmetrically and asymmetrically methylated DNA, it is also suitable to interrogate DNA methylation at CHH sequences, which are vastly abundant in plant genomes.

We demonstrated that DME-qPCR allowed precise quantification of 5mC at any sequence context in a sequence nonspecific manner (Figure 2). Moreover, DME-qPCR successfully distinguished different methylation levels at the regulatory regions of two representative epialleles of plants, *FWA* and *Cnr*, corresponding to their respective phenotypes (Figure 3 and Figure 4). DME-qPCR was able to discern differently methylated alleles of *FWA* in the segregating population (Figure 3D), which may serve as epigenetic molecular markers that can predict flowering time even at seedling stages. Importantly, the 5′ upstream region of *FWA* has no *Hpa* II/*Msp* I recognition sites (Figure 3A,B), and thus, only DME-qPCR can be applicable to any genomic regions of interest without concerns about the availability of recognition sequences of MSREs. In addition, DME-qPCR easily detected changes in DNA methylation level at the DMR of *CNR* during the course of fruit ripening in tomato (Figure 4), and therefore, it will offer valuable information on dynamic changes of DNA methylation in diverse developmental processes in many eukaryotic organisms.

Although we propose DME-qPCR as a promising method for quantitative analysis of DNA methylation, cautions should be taken in several cases. First, incomplete DME reaction may generate false-positive signals during PCR amplification because DME-qPCR depends upon the generation of DNA strand breaks as most MSRE-based PCR assays. Second, DME-qPCR results may not precisely reflect DNA methylation levels at densely methylated regions such as CG repeats because DME 5mC excision is inhibited by nearby strand breaks [22], and excessive SSBs exponentially decrease the template availability. Third, fully and hemi-methylated sites, although the latter is supposedly uncommon at CG dinucleotides in biological samples, cannot be distinguished by DME-qPCR because both generate the same SSB by DME treatment.

Previously, we extensively modified the full-length DME peptide by trimming catalytically unnecessary regions, producing a compact but active fragment [29]. This greatly improved stability and solubility of the protein when purified from *E. coli*, where conventional affinity purification methods were employed, and a batch of purification was stored at −80 °C and used for many reactions. Moreover, most *E. coli* host cells generally used for recombinant protein expression are cytosine methylation-deficient B strain-derivatives (*dcm^-^*) tolerable to DME expression, although DNA methylation-proficient hosts (K strain-derivatives) are sensitive to DME-induced 5mC excision [22]. Therefore, it is fairly affordable to prepare DME proteins in a quantity enough to conduct a small scale DME-qPCR analysis even in budget-tight laboratories.

Although regarded as a gold standard for genome-wide DNA methylation analysis while providing a single-base resolution, BS-Seq still has several intrinsic drawbacks—first, sodium bisulfite converts unmethylated cytosine to thymine reducing sequence complexity; second, C/T single nucleotide polymorphisms (SNPs) present between different samples will be neglected upon bisulfite conversion; third, sodium bisulfite treatment is often destructive towards DNA severely compromising the template quality. Therefore, it is imminent that more precise, cost-effective, and researcher-friendly tools must be developed for better epigenetic analysis replacing current techniques.

The unique enzyme property of DME may be utilized to develop DNA methylation analysis techniques with different principles. For instance, DME-treated genomic fragments will generate distinct fragmentation patterns according to DNA methylation profiles, which may be considered an ‘epigenetic polymorphism’ at the molecular level. Detection of a ‘nick’ created by DME may give positional information where 5mC exists. It is thus plausible that high throughput sequencing technology combined with DME pre-treatment may provide genome-wide DNA methylation information in high resolution. Recent study reported a ‘Nick-seq’ method that was utilized to map diverse DNA modifications at single-nucleotide resolution [30], and the same method can be used for DME-induced nick detection, which would likely offer DNA methylation information comparable to BS-Seq at the genome-wide level. Upon 5mC excision, DME generates unusual 3′ end structures such as 3′-phosphate and 3′-phospor-α, β-unsaturated aldehyde (3′-PUA), which are further processed by AP endonuclease providing 3′-OH for nucleotide extension [28]. We already showed that the trimmed 3′ ends can be extended with a fluorophore-containing nucleotide [28], and such fluorescence-labeled fragments may be used for hybridization-based analysis using microarrays.

Since 5mC is prevalent in many eukaryote genomes serving as a primary epigenetic modification that controls chromatin structure and gene expression, it is important to understand DNA methylation profiles and regulation processes associated with diverse biological phenomena. It is also evident that aberrant DNA methylation is responsible for certain diseases and cancer development in humans [31]. A number of DNA methylation analysis tools are available at the moment, but none of them satisfy all demands from researchers and pharmaceutical industries. Therefore, it is crucial to have a novel method developed serving a better purpose, and we believe that DME has a great versatility and specificity in conceiving innovative methods because DME and its related members are yet the only enzymes that are capable of directly removing 5mC from DNA as a canonical DNA demethylase.

## 4. Materials and Methods

### 4.1. Protein Expression and Purification

DMEΔ677ΔIDR1::lnk with 6xHis and maltose binding protein tag [28,32] was expressed in the *Escherichia coli* Rosetta 2 (DE3) strain (Novagen, Darmstadt, Germany). The protein purification steps were essentially the same as described by Mok et al. [29]. Briefly, the recombinant DME protein was sequentially purified through a HisTrap FF column (GE Healthcare, Chicago, IL, USA) and a HiTrap Heparin HP column (GE Healthcare, Chicago, IL, USA), and gel filtration was performed on a HiLoad 16/60 Superdex 200-pg column (GE Healthcare, Chicago, IL, USA). The final eluted fractions were concentrated and aliquoted with 50% glycerol.

### 4.2. DME Treatment and Bal-31 Digestion Followed by PCR on Methylated DNA Fragments

Yeast genomic DNA was extracted using a standard protocol [33]. In total, 10 µg of yeast gDNA was in vitro methylated using 10 units of *Hpa* II methyltransferase (NEB, Ipswich, MA, USA) at 37 °C for 2 h. The methylated yeast gDNA (2 µg) was treated with 200 ng of DMEΔN677ΔIDR1::lnk at 37 °C for 2 h in glycosylase reaction buffer (10 mM Tris-HCl, pH 7.4, 50 mM NaCl, 0.5 mM dithiothreitol (DTT), 200 μg/mL BSA) and the reaction was inactivated at 65 °C for 15 min. For DBS formation, the reaction was incubated with 1 unit of the single-strand specific nuclease BAL-31 (NEB, Ipswich, MA, USA) at 30 °C for 1 h in BAL-31 reaction buffer (NEB, Ipswich, MA, USA) (600 mM NaCl, 12 mM CaCl_2_, 12 mM MgCl_2_, 20 mM Tris-HCl pH 8, 1 mM EDTA). The reactions with DME and BAL-31 were carried out sequentially in a single tube. The PCR primers for amplification of unmethylated and methylated regions are listed in Appendix A. After an initial denaturation at 95 °C for 5 min, the thermal cycling conditions were as following: 95 °C for 30 s, 56 °C for 30 s, and 72 °C for 60 s. The PCR cycles were adjusted between 20 and 26 for the quantification.

### 4.3. Comparison of DME Treatment and McrBC Digestion

Unmethylated pUC19 plasmid was purified from the methylation-deficient *E. coli* strain JM110. Next, 1 μg of the pUC19 plasmid was methylated using either 10 units of the CG methyltransferase M.*Sss* I (NEB, Ipswich, MA, USA) or eight units of *Hpa* II methyltransferase. For representing various DNA methylation levels, unmethylated and M.*Sss* I-methylated plasmids were mixed at different ratios. pUC19 plasmid (20 ng) was incubated either with 400 ng of DMEΔN677ΔIDR1::lnk in glycosylase reaction buffer or with 20 units of McrBC (NEB, Ipswich, MA, USA) and 1 mM GTP in NEBuffer 2 (10 mM Tris-HCl, pH 7.9, 50 mM NaCl, 10 mM MgCl_2_, 1 mM DTT) at 37 °C for 1 h, and the reactions were inactivated at 65 °C for 15 min. An untreated control, neither treated with DME or McrBC, was prepared following the same procedure. The reaction products were diluted 1:2000 for subsequent tests. Quantitative PCR (qPCR) was performed using the Rotor-Gene Q system (Qiagen, Hilden, Germany) with SYBR green Q-master mix (Genet Bio, Daejeon, Korea). The PCR primers for amplification of the UMR, HMR, and LMR are listed in Appendix A. qPCR was performed at 95 °C for 10 min, followed by 40 cycles of 95 °C for 10 s, 60 °C for 15 s, and 72 °C for 35 s. The relative amplification was calculated using the ΔΔCt method. To calculate ΔCt, the Ct of the unmethylated control region was subtracted from the Ct of the target region. To obtain ΔΔCt, the ΔCt value of the untreated sample was subtracted from the ΔCt value of the enzyme-treated sample. The relative amplification (RA) value was determined using the equation RA = 2^(−ΔΔCt)^, and the standard deviations were calculated after 2^−Ct^ transformation [34].

### 4.4. Plant Materials

*Arabidopsis thaliana* ecotypes Columbia (Col-0), Landsberg *erecta* (L*er*), and *fwa-1* in the L*er* background were used in this study [35]. The seeds were sterilized and stratified at 4 °C for 2 days in the dark, grown on half-strength Murashige–Skoog (MS) media agar plates, and transplanted to soil in a growth chamber with 16 h per day of fluorescent light (20 ± 5 μmol m^−2^ s^−1^) at 22 ± 1 °C and 70 ± 5% relative humidity. The F_2_ progenies were generated by crossing Col-0 and *fwa-1* (L*er*) followed by self-crossing. Flowering time was measured by counting the number of days from sowing and rosette leaf number until bolting.

Wild-type tomatoes (*Solanum lycopersicum* cultivar Ailsa Craig) and the corresponding *Cnr* mutant were also used in this study [9]. The seeds were sterilized and grown in a growth chamber for 1 month. They were transplanted to soil and grown in greenhouse conditions (12 h with supplemental lighting at 25 °C and 12 h at 20 °C) with regular additions of N-P-K fertilizer (Hyponex). Tomato fruit pericarp tissues were harvested at 17, 39, 42, and 52 d.p.a.

### 4.5. CAPS Analysis

Genomic DNA was extracted from *Arabidopsis* leaves using CTAB extraction method. PCR products were amplified with CAPS primers and then digested with 20 units of Taq^α^I restriction enzyme for 4 h at 65 °C. Gel electrophoresis was conducted on a 2.5% agarose gel at 25 V. CAPS primers were designed to contain SNPs between Col-0 and L*er* within the Taq^α^I restriction site (Appendix A).

### 4.6. Reverse Transcriptase-qPCR

Total RNA was extracted from fruit pericarp tissues, which were harvested at 17, 39, 42, and 52 d.p.a., of wild type (Ailsa Craig) and *Cnr* mutant using the TRIzol (Ambion, Waltham, MA, USA). Next, 1 μg of total RNA was reverse transcribed into cDNA by using the QuantiTect Reverse Transcription Kit (Qiagen, Hilden, Germany) with manufacturer’s instruction. The PCR primers for amplification of *CAC*, *TIP41,* and *Cnr* genes are listed in Appendix A. The average Ct value of *CAC* and *TIP41* was used as a control. The relative amplification was calculated using the ΔΔ Ct method [34].

### 4.7. Locus-Specific Bisulfite Sequencing

Bisulfite conversion of genomic DNA was performed using the EpiTect Bisulfite kit (Qiagen, Hilden, Germany) according to the manufacturer’s protocols. The degenerative primers for PCR amplification are listed in Appendix A. The PCR products were cloned into the TA vector (Real Biotech Corporation, Banqiao City, Taipei), and individual clones were sequenced. DNA methylation in each context (CG, CHG, and CHH) was analyzed using CyMATE software [36].

### 4.8. DME-qPCR

Approximately 500 ng of gDNA was incubated with 600 ng of DMEΔN677ΔIDR1::lnk in glycosylase reaction buffer at 37 °C for 2 h in 20 µL. Following heat-inactivation at 65 °C for 15 min, 50 ng of DME-treated DNA was subjected to qPCR amplification. An untreated control, not treated with DME, was prepared following the same procedure. The remainder of the qPCR procedures were performed as described above. The relative amplification of target region was normalized to the unmethylated region. The DME-qPCR primers are listed in Appendix A.

## Figures and Tables

**Figure 1 ijms-22-01072-f001:**
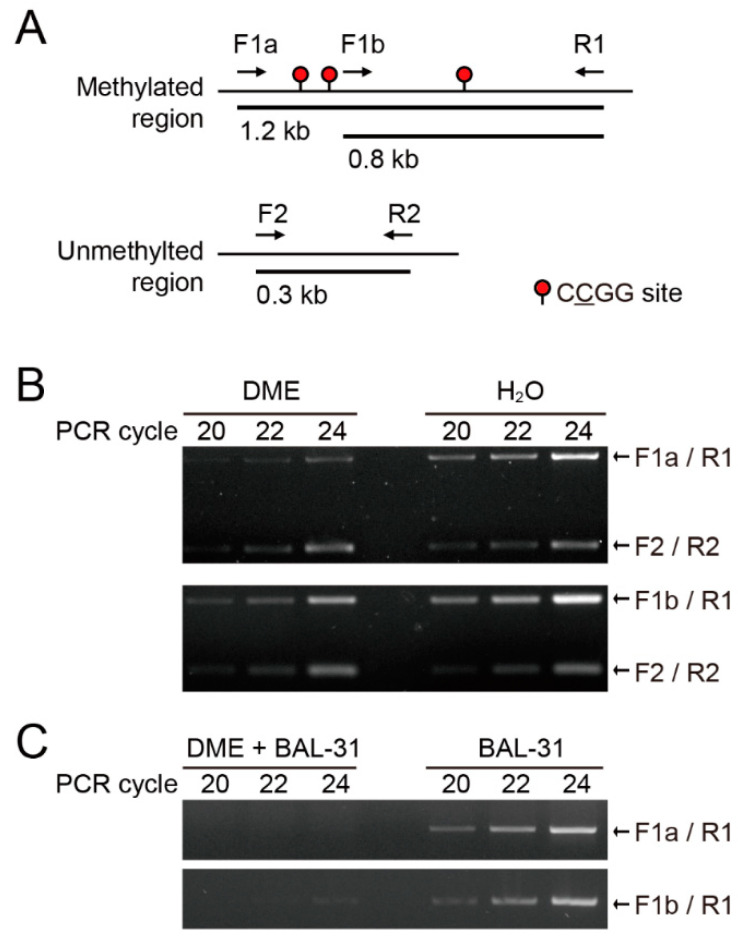
Quantitative analysis of DNA methylation by DEMETER (DME) treatment and PCR amplification. (**A**) Schematic diagrams of differentially methylated regions. Yeast genomic DNA was in vitro methylated with *Hpa* II methyltransferase for CCGG methylation. Methylated regions with a single or three CCGG sites were subjected to DME treatment and PCR amplification with F1a/R1 and F1b/R1 primer pairs, respectively. Unmethylated control region was amplified with F2/R2 primer pairs. (**B**) DME-induced SSBs decreases PCR amplification in proportion to DNA methylation levels. *Hpa* II-methylated yeast genomic DNA was treated with DME and subjected to multiplex PCR amplification. Primer pairs are denoted to the right of the panel. (**C**) DME and BAL-31 treatment induces DSBs at methylated regions. DSB formation by DME and BAL-31 treatment prevents PCR amplification regardless of the number of methylated sites.

**Figure 2 ijms-22-01072-f002:**
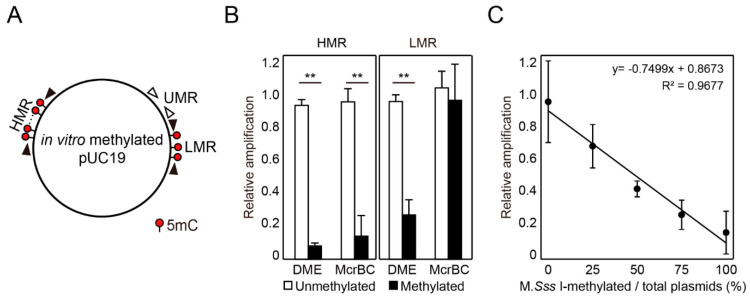
DME-qPCR analysis for quantification of the level of DNA methylation. (**A**) Schematic representation of the pUC19 plasmid with DMRs. For the heavily and low methylated regions (HMR and LMR), the pUC19 plasmid was in vitro methylated with M.*Sss* I and *Hpa* II methyltransferases, respectively, producing differentially methylated HMR (20 5mC sites) and LMR (three 5mC sites) regions. Differentially methylated HMR and LMR, along with unmethylated UMR as a control, were subjected to qPCR analysis with corresponding primers. HMR, heavily methylated region; LMR, low methylated region; UMR, unmethylated region. (**B**) Comparison between DME and McrBC treatments for quantification of DNA methylation. Following the enzyme treatment, HMR, LMR, and UMR were subjected to qPCR analysis with corresponding primer pairs. Relative amplification was calculated using the ΔΔCt method for unmethylated (white bar) and methylated (black bar) pUC19 plasmids normalized with the value obtained from UMR. (**C**) Relative amplifications for different ratios of methylated DNA by DME-qPCR. The percentage (x-axis) indicates the ratio of M.*Sss* I-methylated plasmids to the total plasmids in the reaction. Relative amplification values were obtained for HMR from DME-qPCR, and a regression line fitted to the observed data. The data represent the mean ± S.D. for three replicates (**B**,**C**). Asterisks indicate statistically significant differences between the unmethylated and methylated plasmids (**: *p* < 0.01; Student’s *t*-test).

**Figure 3 ijms-22-01072-f003:**
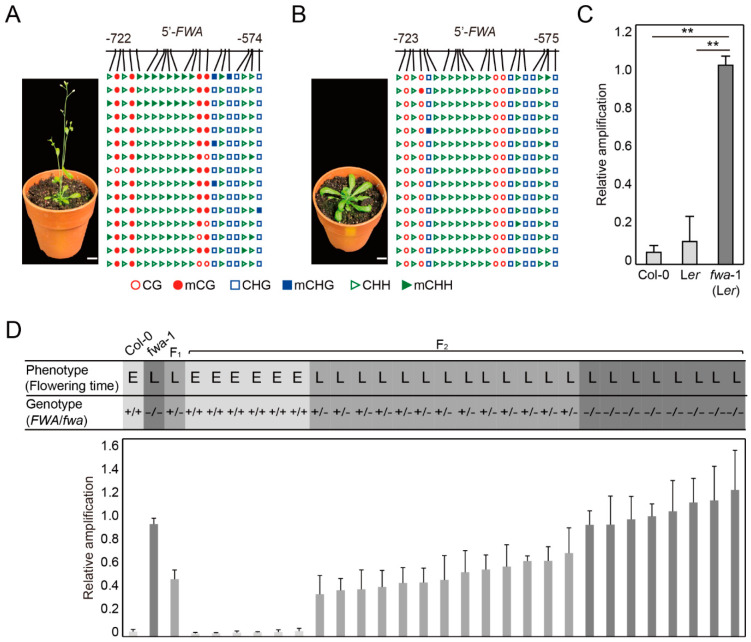
DME-qPCR for quantitative DNA methylation analysis for *FWA* alleles in *Arabidopsis*. (**A**,**B**) Flowering time phenotypes and DNA methylation profiles of Col-0 wild type (**A**) and *fwa-1* mutant in the L*er* background (**B**). The 5′-*FWA* represents an upstream region of the translation start site of *FWA*. DNA methylation patterns were analyzed using bisulfite sequencing. Genomic DNA isolated from individual plants was subjected to bisulfite sequencing analysis. Scale bar = 10 mm. (**C**) DME-qPCR analysis for wild type (Col-0 and L*er*) and *fwa-1* mutant plants. Relative amplifications were calculated using the ΔΔCt method with the value for unmethylated *ASA1* gene as a control. Asterisks indicate statistically significant differences between wild type and *fwa-1* (**: *p* < 0.01; Student’s *t*-test). (**D**) DME-qPCR analysis for *FWA* alleles in the F_2_ population. F_2_ population was generated by selfing F_1_ obtained from a cross between wild type (Col-0) and *fwa-1* (L*er*). ‘E’ and ‘L’ indicate early and late flowering phenotype, respectively. ‘+’ and ‘−’ indicate wild type (Col-0) and *fwa-1* (L*er*) genotypes, respectively, at the *FWA* locus analyzed with the CAPS marker. Light, medium and dark gray colors indicate genotypes +/+, +/− and −/− at the *FWA* locus. DME-qPCR amplification values were calculated relative to *ASA1*. The data represent the mean ± S.D. of three replicates (**C**,**D**).

**Figure 4 ijms-22-01072-f004:**
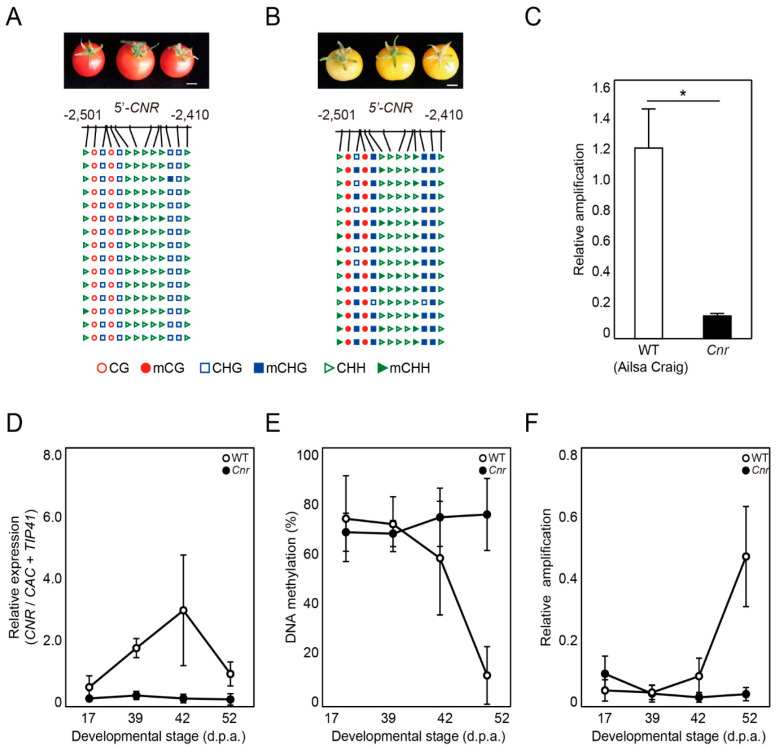
DME-qPCR for quantitative DNA methylation analysis for *Cnr* alleles during fruit ripening in tomato. (**A**,**B**) Mature fruit colors and DNA methylation profiles at 2.4–2.5 kb upstream of the translation start site of *CNR* gene in wild type Ailsa Craig (**A**) and *Cnr* mutant (**B**), respectively. DNA methylation profiles were analyzed by bisulfite sequencing. Genomic DNA isolated from individual plants was subjected to bisulfite sequencing analysis. Scale bar = 10 mm. (**C**) DME-qPCR analysis for *CNR* in wild type Ailsa Craig and *Cnr* mutant leaves. DME-qPCR amplification values were calculated relative to unmethylated *ATP1* gene. Asterisks indicate statistically significant differences between WT and *Cnr* (*: *p* < 0.05; Student’s *t*-test). (**D**) Expression of *CNR* in wild-type and *Cnr* mutant tomato fruits during ripening. *CNR* expression was determined by RT-qPCR in fruit pericarp at 17, 39, 42, and 53 d.p.a. Expression levels were relative to combined expressions of *CAC* and *TIP41* [26]. (**E**) DNA methylation levels of *CNR* in wild-type and *Cnr* mutant tomato fruits during ripening. DNA methylation levels at the DMR 0.7 kb upstream of the translation start site of *Cnr* gene were determined by bisulfite sequencing in pericarp tissues of wild type Ailsa Craig and *Cnr* mutant at 17, 39, 42, and 52 d.p.a. (**F**) DME-qPCR analysis for detection of DNA methylation changes at *CNR* during fruit ripening. Changes in DNA methylation level at the −0.7 kb DMR of *Cnr* gene were examined by DME-qPCR. DME-qPCR amplification values were calculated relative to *Actin* (Solyc01g066800). The data represent the mean ± S.D. of three replicates (**C**,**D**,**F**).

## Data Availability

The data presented in this study are available on request from the corresponding author.

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
