# Peer review of "Application of 5-Methylcytosine DNA Glycosylase to the Quantitative Analysis of DNA Methylation"

_ijms, 2021, doi:10.3390/ijms22031072_

Round 1

Reviewer 1 Report

In this study, Woo Lee Choi et al present an innovative approach to quantifying methylation levels at a sequence of interest, using a plant 5mC DNA glycosylase enzyme. The approach is novel, thought-provoking, and in my view should be accepted for publication, so that the wider field can attempt this method and see if it useful. The experiments in the paper are well designed and performed, and the manuscript is clearly written. While the technique may not catch on (see some concerns below), I believe that a wider readership is a better mechanism than peer review for assessing if this method provides value to the field.

Current methods for examining methylation levels have several drawbacks. While bisulfite PCR provides high quality data with single-cytosine resolution, it is time-consuming, potentially costly, and requires large quantities of input DNA. Most of the currently-used quicker/easier methods based on methylation-sensitive restriction digestion (e.g. Chop-qPCR) are in my view, sufficiently flawed as to not be suitable for publication in many cases – yet they persist in the field due to the drawbacks of bisulfite PCR. If found to be easy to perform and replicable, the approach outlined in this paper could provide a more reliable method of quantifying methylation that could complement (although likely not replace) bisulfite PCR. This method may also provide more reliable data for labs working on DNA methylation with a tight budget.  

Before this work is published, I believe the authors should include honest discussion into the potential drawbacks of their method, so that these potential pitfalls are clear to readers. Some points for discussion that would be valuable include:

  • Is purifying DME for this experiment simple, affordable and easy to replicate without training?
  • Is purifying DME for this experiment time-efficient? For example, do researchers need to purify DME each time they perform the experiment, or can a lab reliably stock-up once a year?
  • If a single cytosine or CG dinucleotide is methylated in all cells in one condition versus a control, it could result in substantially lower amplification (as is visible in Fig. 2B LMR), even though the methylation difference is unlikely to be biologically significant. Bisulfite PCR remains the gold standard in this regard, as it discerns both inter-strand and inter-strand variation at single-cytosine resolution. The authors should either present data on how a single CG dinucleotide can affect the results, or discuss this point fully in the discussion. Spontaneous “epimutation” of CG dinucleotides is common and well-understood, and could lead to false positives in experiments using this approach.
  • The experiment is limited by the amplicon size of efficient qPCR. Bisulfite PCR is therefore able to assay longer amplicons in some cases.

Author Response

In this study, Woo Lee Choi et al present an innovative approach to quantifying methylation levels at a sequence of interest, using a plant 5mC DNA glycosylase enzyme. The approach is novel, thought-provoking, and in my view should be accepted for publication, so that the wider field can attempt this method and see if it useful. The experiments in the paper are well designed and performed, and the manuscript is clearly written. While the technique may not catch on (see some concerns below), I believe that a wider readership is a better mechanism than peer review for assessing if this method provides value to the field.

Current methods for examining methylation levels have several drawbacks. While bisulfite PCR provides high quality data with single-cytosine resolution, it is time-consuming, potentially costly, and requires large quantities of input DNA. Most of the currently-used quicker/easier methods based on methylation-sensitive restriction digestion (e.g. Chop-qPCR) are in my view, sufficiently flawed as to not be suitable for publication in many cases – yet they persist in the field due to the drawbacks of bisulfite PCR. If found to be easy to perform and replicable, the approach outlined in this paper could provide a more reliable method of quantifying methylation that could complement (although likely not replace) bisulfite PCR. This method may also provide more reliable data for labs working on DNA methylation with a tight budget.

Before this work is published, I believe the authors should include honest discussion into the potential drawbacks of their method, so that these potential pitfalls are clear to readers.

Thanks for the suggestion. We added a paragraph describing potential pitfalls of the DME-qPCR method in Discussion (L317-326) as below:

“Although we propose DME-qPCR as a promising method for quantitative analysis of DNA methylation, cautions should be taken in several cases. First, incomplete DME reaction may generate false-positive signals during PCR amplification because DME-qPCR depends upon the generation of DNA strand breaks as most MSRE-based PCR assays. Second, DME-qPCR results may not precisely reflect DNA methylation levels at densely methylated regions such as CG repeats because DME 5mC excision is inhibited by nearby strand breaks [22], and excessive SSBs exponentially decrease the template availability. Third, fully and hemi-methylated sites, although the latter is supposedly uncommon at CG dinucleotides in biological samples, cannot be distinguished by DME-qPCR because both generate the same SSB by DME treatment.”

Some points for discussion that would be valuable include:

Is purifying DME for this experiment simple, affordable and easy to replicate without training?

Yes, it is quite straightforward to purify DME proteins expressed in E. coli. General expression and purification steps were used, and the details were previously reported by Mok et al. (2010). Briefly, E. coli Rosetta 2 (DE3) strains were used as an expression host, and protein expression was induced with 0.1 mM IPTG at 28 °C overnight. Cells were lysed by sonication, and affinity purification was performed over HisTrap and HiTrap Heparin columns followed by size separation over the Superdex column. Approximately 2-3 mg of DME protein was typically obtained from a single batch of purification and stored at -80 °C until use. Such affordability is briefly described in Discussion (L327-336) as below:

“Previously, we extensively modified the full-length DME peptide by trimming catalytically unnecessary regions, producing a compact but active fragment [32]. This greatly improved stability and solubility of the protein when purified from E. coli, where conventional affinity purification methods were employed, and a batch of purification was stored at -80 °C and used for many reactions. Moreover, most E. coli host cells generally used for recombinant protein expression are cytosine methylation-deficient B strain-derivatives (dcm-) tolerable to DME expression, although DNA methylation-proficient hosts (K strain-derivatives) are sensitive to DME-induced 5mC excision [22]. Therefore, it is fairly affordable to prepare DME proteins in a quantity enough to conduct a small scale DME-qPCR analysis even in budget-tight laboratories.”

Is purifying DME for this experiment time-efficient? For example, do researchers need to purify DME each time they perform the experiment, or can a lab reliably stock-up once a year?

Yes. As described above, DME can be prepared in a week from transformation to purification using conventional methods. Purified DME can be stored at -80 °C in 50% glycerol for more than a year. This part is added in Discussion. Please see the above paragraph. We hope that commercialization of DME will further facilitate the use of DME in molecular biology and diagnostics as well.

If a single cytosine or CG dinucleotide is methylated in all cells in one condition versus a control, it could result in substantially lower amplification (as is visible in Fig. 2B LMR), even though the methylation difference is unlikely to be biologically significant. Bisulfite PCR remains the gold standard in this regard, as it discerns both inter-strand and inter-strand variation at single-cytosine resolution. The authors should either present data on how a single CG dinucleotide can affect the results, or discuss this point fully in the discussion. Spontaneous “epimutation” of CG dinucleotides is common and well-understood, and could lead to false positives in experiments using this approach.

We much appreciate the valuable comments. DME-qPCR cannot discern strand-specific or cell-specific differences, and thus, a thorough investigation still requires bisulfite conversion. Most enzyme-based methods without sequence verification have similar drawbacks. In addition, intrinsic limitations when addressing the effect of hemi-methylation at CG dinucleotides and their abundance were described in a newly added paragraph (L317-326), and we hope that readers will easily figure out possible cases of ‘false positives’ in results interpretation.

The experiment is limited by the amplicon size of efficient qPCR. Bisulfite PCR is therefore able to assay longer amplicons in some cases.

Yes, we agree. In practice, DME-qPCR is more suitable for local DNA methylation profiling, and a ‘DME-seq’ adopting a recently developed ‘Nick-seq’ method may overcome such limitations while providing higher coverage and resolution. This possibility was briefly mentioned in Discussion.

Reviewer 2 Report

Overall this technical manuscript presenting the use of recombinant DEMETER domains fusion to assess the methylation levels at specific loci is interesting, well written, clear and simple. I think it is of interest for a large public, as bisulfite-sequencing of whole genomes is not always relevant for all applications.The introduction is well written : no specific question raised

Using a semi quantitative approach, the authors first show that recombinant DME induces single-strand DNA breaks proportional to DNA methylation density. This part may look a bit old fashioned as compared to a more quantitative approach but I like its simplicity and beauty. the authors then move on to DME-qPCR approach and proves that it is possible to reach quantitative DNA methylation analysis. In this part (Fig2B), the comparison with McrBC is very significant and convincing.

Then the authors are able to quantify endogenous methylation level for two loci : one in Arabidopsis, and one in Tomato. They show that recombinant DME-qPCR distinguishes DNA methylation levels at the FWA gene in wild type and late flowering mutants in Arabidopsis, and also faithfully detects changes in DNA methylation levels at the CNR gene during fruit ripening in tomato. This data is sound and clearly stated.

Minor comments: relative amplification was measured using the 2 Ct. This method is perfectly valid, however it would be nice to mention primer efficiency, adding it in the suppl. table describing primers.

Figure  2C: could the same result be obtained with LMR region amplification? This is the type of region where the technique could be particularly interesting to use.

“Local bisulfite sequencing showed hypermethylation in the promoter region, particularly at CG sequences, of the FWA gene in early-flowering wild type (Figure 3A) but hypomethylation in late flowering fwa mutants (Figure 3B). “

The authors did not clearly mention that (probably) individual plants were used for local bisulfite sequencing.

Figure 3C : same question : was each biological replicate done on a pool or single individuals?

“It was reported that two differentially methylated regions  (DMRs) located at 0.7 kb and 1.9 kb upstream of the translation start site of CNR underwent gradual loss of DNA methylation during fruit ripening for transcriptional activation [23].”  The use of “for” sounds akward.  Maybe state this sentence more clearly (“allowing fruit maturation” or something like that).

The discussion, also well written, raises no particular question.

Author Response

Overall this technical manuscript presenting the use of recombinant DEMETER domains fusion to assess the methylation levels at specific loci is interesting, well written, clear and simple. I think it is of interest for a large public, as bisulfite-sequencing of whole genomes is not always relevant for all applications. The introduction is well written : no specific question raised

Using a semi quantitative approach, the authors first show that recombinant DME induces single-strand DNA breaks proportional to DNA methylation density. This part may look a bit old fashioned as compared to a more quantitative approach but I like its simplicity and beauty. the authors then move on to DME-qPCR approach and proves that it is possible to reach quantitative DNA methylation analysis. In this part (Fig2B), the comparison with McrBC is very significant and convincing.

Then the authors are able to quantify endogenous methylation level for two loci : one in Arabidopsis, and one in Tomato. They show that recombinant DME-qPCR distinguishes DNA methylation levels at the FWA gene in wild type and late flowering mutants in Arabidopsis, and also faithfully detects changes in DNA methylation levels at the CNR gene during fruit ripening in tomato. This data is sound and clearly stated.

Minor comments: relative amplification was measured using the ΔΔCt. This method is perfectly valid, however it would be nice to mention primer efficiency, adding it in the suppl. table describing primers.

> We much appreciate the comment. We did not test the primer efficiency using standard curves with different pairs of primers. The primer sets were empirically tested and the best ones chosen. If information on the primer efficiency is necessary, we are willing to perform the test. However, we cannot finish in time within 5 days of revision at this moment.

Figure 2C: could the same result be obtained with LMR region amplification? This is the type of region where the technique could be particularly interesting to use.

> We believe the similar results will be obtained with LMR. However, we presume that inclination of the fitted line (Fig. 2C) may be slightly lowered because the difference in relative amplification between unmethylated and methylated templates is smaller (right panel in Fig. 2B). In other words, the DME-treated region with three CG methylation sites (LMR) would statistically have a quarter of strands intact without a strand break, and thus more templates will be available for PCR amplification than HMR (with 20 CG sites).

“Local bisulfite sequencing showed hypermethylation in the promoter region, particularly at CG sequences, of the FWA gene in early-flowering wild type (Figure 3A) but hypomethylation in late flowering fwa mutants (Figure 3B). “

The authors did not clearly mention that (probably) individual plants were used for local bisulfite sequencing.

> Individual plants were used for local bisulfite sequencing analysis. This information was added in the figure legend (L221) as below:
“Genomic DNA isolated from individual plants was subjected to bisulfite sequencing analysis.”

Figure 3C : same question : was each biological replicate done on a pool or single individuals?

> Individual plants were also used for local bisulfite sequencing analysis. This information was added in the figure legend (L263).

“It was reported that two differentially methylated regions (DMRs) located at 0.7 kb and 1.9 kb upstream of the translation start site of CNR underwent gradual loss of DNA methylation during fruit ripening for transcriptional activation [23].” The use of “for” sounds akward. Maybe state this sentence more clearly (“allowing fruit maturation” or something like that).

> We replaced “for” with “allowing” as suggested (L242).

The discussion, also well written, raises no particular question.